# Seemingly Simple Planning Problems are Computationally Challenging: The Countdown Game

## Abstract

There is a broad consensus that the inability to form long-term plans is one of the key limitations of current foundational models and agents. However, the existing planning benchmarks remain woefully inadequate to truly measure their planning capabilities. Most existing benchmarks either focus on loosely defined tasks like travel planning or end up leveraging existing domains and problems from international planning competitions. While the former tasks are hard to formalize and verify, the latter were specifically designed to test and challenge the weaknesses of existing automated planners. To address these shortcomings, we propose a *procedure* for creating a planning benchmark centered around the game called *Countdown*, where a player is expected to form a target number from a list of input numbers through arithmetic operations. We discuss how this problem meets many of the desiderata associated with an ideal benchmark for planning capabilities evaluation. Specifically, the domain allows for an intuitive, natural language description for each problem instance, it is computationally challenging (NP-complete), and the instance space is rich enough that we do not have to worry about memorization. We perform an extensive theoretical analysis, establishing the computational complexity result and demonstrate the advantage of our instance generation procedure over public benchmarks. We evaluate a variety of existing LLM-assisted planning methods on instances generated using our procedure. Our results show that, unlike other domains like 24 Game (a special case of Countdown), our proposed dynamic benchmark remains extremely challenging for existing LLM-based approaches.

## 1 Introduction

The inability to come up with long-term sequential plans remains a core hurdle to using foundational models and large language models (LLMs) to create highly autonomous agents. Thus, benchmarking the planning ability of such models and agents is of paramount importance. Surprisingly, the current set of approaches to measuring planning capabilities is quite limited. Looking at the current landscape, one can easily recognize two main trends. First, a set of benchmarks that focus on easy-to-specify and intuitive but fuzzy planning tasks like travel-planning (Xie et al., 2024; Zheng et al., 2024). Unfortunately, such domains are hard to formalize, making a rigorous evaluation of planning capabilities nearly impossible to achieve. Second, a set of benchmarks that builds off of international planning competition (IPC) domains (Bacchus, 2001) that were originally designed to evaluate the performance of automated planners (Valmeekam et al., 2023; Kokel et al., 2025). While this category of benchmarks could, in theory, offer more diversity and the ability to perform systematic evaluation, the specific domains and problems were designed to challenge the strengths and weaknesses of planners that were popular at the time of these competitions. Additionally, these planning domains may not be easy to specify in intuitive natural language prompts (Stein et al., 2025).

Consequently, LLM researchers looked at logical puzzles for benchmark domains. Among them, the 24 Game, popularized by ToT (Yao et al., 2023), and widely used since. While easy to describe in natural language, the puzzle is restricted in size, with a state space of around 4500 states (Katz et al., 2024). While several methods show significant performance on this dataset, the benchmark used by most methods consists of instances scraped from the internet (Yao et al., 2023), raising concerns

of data contamination. An alternative that was recently considered is the game called **Countdown**[1] (Gandhi et al., 2024). In this game, a player receives a list of numbers and is asked to form a given target number through a sequence of arithmetic operations. This is a strict generalization of the 24 Game, which only considers the target number $24$ and input of size $4$. While the game becomes more popular as a benchmark (Stojanovski et al., 2025), there has been surprisingly little effort to understand its nature and complexity. Such a lack of clear understanding of the computational nature of the problem could lead to misinterpretation of the experimental results and possibly overestimating the true planning capabilities of the tested methods. To exemplify, a good generalization capability may be claimed when observing non-decreasing performance as instances grow in size. This, however, is true only if the problem hardness grows monotonically with instance size in that range. This assumption turns out not to hold in Countdown, irrespective of the instance generation method. We alleviate this gap in understanding of the Countdown by providing a rigorous and thorough analysis of the problem. More specifically, our contributions are as follows:

1. We establish that Countdown is an NP-complete problem.

2. We provide an approach for generating challenging Countdown problem instances and compare it to existing approaches in the literature.

3. We create a novel formulation of Countdown in a planning language PDDL, allowing us to leverage existing numeric planners as a baseline.

4. We conduct a rigorous experimental evaluation of a representative collection of existing LLM-assisted planning methods. We show that the AutoToS method (Cao et al., 2024), which uses LLMs to generate a symbolic solver, performs well on the tested collection, surpassing the domain-independent planner baseline. Our experiments reveal two surprising results.

   • We discover an interesting phenomena in Countdown, two phase transitions as instance size grows. The first one is natural, from easy to hard instances, while the second one is surprising, from hard to easy instances.
   • We find the famous LLM-based methods (Wei et al., 2022; Yao et al., 2023) to struggle with the instances in the tested collection, even with instances of smallest size. The performance of these methods on our dataset is dramatically worse than on the static dataset they were originally tested on, hinting that the reported in the literature performance levels may have been due to memorization.

5. We perform an analysis of errors generated by the LLM-based planners on the domain.

## 2 PLANNING BENCHMARK DESIDERATA

We start by listing a few desired properties for a successful benchmark of planning abilities.

   • The problem should be sequential in nature, the order in which the actions need to be performed should matter.
   • It should have a well defined action and state space.
   • The problem should be of a non-trivial complexity.
   • It should have a precise yet concise natural language description, including initial state, goal, and task dynamics.
   • Must have sound validators for candidate solutions.
   • It should have a large instance space and a dynamic generation procedure, thus allowing for the avoidance of memorization concerns.

We will show the Countdown problem meet these criteria.

---

[1]It is loosely (Colton, 2014) based on a popular French game show *Des chiffres et des lettres* and its British variant under the name *Countdown*.

## 3  Background

We consider planning tasks that are given by their transition system $\Pi = \langle S, A, T, s_0, S_* \rangle$, where $S$ is a finite set of *states*, with $s_0 \in S$ being the *initial* state and $S_* \subseteq S$ being the set of *goal states*. The set $A$ is a finite set of *actions*. The *transition relation* $T \subseteq S \times A \times S$ is deterministic, i.e. for every state $s$ and action $a$, there is at most one $s'$ with $(s, a, s') \in T$. If there is such an $s'$, we say that $a$ is *applicable* in $s$ and that $s'$ is the successor state achieved by applying $a$ in $s$. A *plan* $\pi$ is a sequence of actions that is consecutively applicable in the initial state $s_0$ and where the final state is a goal state.

## 4  The Countdown

We start with the formal definition of the Countdown problem. First, we will restrict our attention here to the set of arithmetic operations $O = \{+, -, *, /\}$. For each operation $o \in O$ and two non-negative rational numbers $x, y$, we will denote the outcome of an arithmetic operation on these numbers as $o(x, y)$. Now with these notations in place, we are ready to define the countdown problem formally.

**Definition 1.** *A **Countdown** problem is defined by a tuple of the form $\mathcal{C} = \langle I_1, O, \tau \rangle$, where* input $I_1$ *is a multi-set of $n$ non-negative integers, i.e, $\forall x \in I_1$, $x \in \mathbb{N}$, operators $O$ is the set of arithmetic operators and* target $\tau$ *is a non-negative integer $\tau \in \mathbb{N}$. The solution to a countdown problem consists of a sequence of triplets of the form $\Theta = \langle \langle x_1, o_1, y_1 \rangle, \ldots, \langle x_{n-1}, o_{n-1}, y_{n-1} \rangle \rangle$, such that*

(i) *for $1 \le i < n$, $o_i \in O$,*

(ii) *for $1 \le i < n$, $\{x_i, y_i\} \subseteq I_i$ and $I_{i+1} = I_i \setminus \{x_i, y_i\} \cup \{o_i(x_i, y_i)\}$, and*

(iii) *$I_n = \{\tau\}$.*

We now show how a Countdown problem $\mathcal{C} = \langle I_1, O, \tau \rangle$ over input size $n$ induces a transition system $\Pi = \langle S, A, T, s_0, S_* \rangle$. First, let us observe that we can over-approximate a set of all rational numbers obtainable from the input in under $n$ steps: Let $\bar{I}_1 \subset \mathbb{N}$ be the set of integer numbers in $I_1$ and $\bar{I}_{i+1} = \{o(x, y) \mid x, y \in \bar{I}_i, o \in O\} \cup \bar{I}_i$. The set $\bar{I}_n$ of all possible reachable numbers in less than $n$ steps is denoted by $\bar{I}$. Clearly, the size of $\bar{I}$ is finite for a finite $n$. Given the set $\bar{I}$, we can now define the set of states $S$, as all multi-sets of size up to $n$ of elements from $\bar{I} \cup \{\tau\}$. The initial state $s_0$ is $I_1$ and the set of goal states $S_*$ is $\{\{\tau\}\}$. The set of all actions is $A = \{\langle o, x, y \rangle \mid x, y \in \bar{I}, o \in O\}$. The transition relation $T$ is defined as follows. For a multi-set $s \in S$, and an action $a = \langle o, x, y \rangle \in A$, $a$ is applicable in $s$ if and only if $\{x, y\}$ is a subset of $s$. In such case, $(s, a, s') \in T$ for $s' = s \setminus \{x, y\} \cup \{o(x, y)\}$.

### 4.1  State Space Size

One can think of the state space $\mathbb{S}$ of the problem as the set of states reachable from the initial state $s_0$ through transitions in $T$. The number of applicable actions (a.k.a. branching factor) in a state $s$ of size $k$ for $k > 1$ is at most $b_k = k * (k-1) * 3$. If we start with a state of size $n$, then the first layer has 1 state, the second layer has $b_n$ states, the third layer has $b_n * b_{n-1}$, and the last layer (layer $n$) has $\prod_{i=2}^{n} b_i$ states. So, layer $j$, $j \ge 2$ has at most $L_j$ states, where $L_j$ is as follows.

$$L_j = \prod_{i=n+2-j}^{n} b_i = \prod_{i=n+2-j}^{n} 3i(i-1) = \frac{3^{j-1} n! (n-1)!}{(n-j)!(n+1-j)!}$$

and the total number of states is therefore bounded by

$$\sum_{j=1}^{n} L_j = \sum_{j=1}^{n} \frac{3^{j-1} n! (n-1)!}{(n-j)!(n+1-j)!}.$$

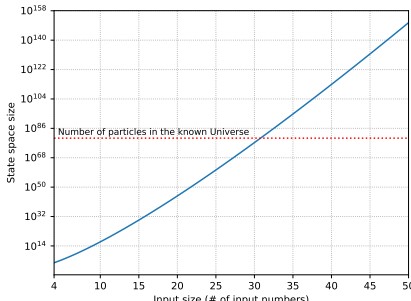

Figure 1: The state space size for the Countdown problem.

Figure 1 shows the state space size (log scale) as a function of state size.

## 4.2 Complexity Analysis

We now analyze the computational complexity of solving the Countdown problem. We start with some useful results from the literature on related problems.

**Definition 2** (3-PP). *3-Partition Problem - For a given multiset of integers $X = \{x_1, ..., x_n\}$ with $n = 3m$ and the sum equal to $mT$, can you partition them into $m$ subsets of $3$ elements each, $X_1, \ldots, X_m$, such that the sum of elements in each $X_i$ is equal to $T$?*

**Lemma 1.** *3-PP is strongly NP-complete.*

The result is by Garey & Johnson (1979). We now move to the intermediate result that helps us showing the main theorem.

**Lemma 2.** *Let $X = \{x_1, ..., x_n\}$ for $n \geq 2$ be a multiset of natural numbers and $B > n$ some natural number. There exists no natural number $y$ such that $\sum_{i=1}^n B^{x_i} = B^y$.*

*Proof.* Let us view every natural number in base $B$. The number $B^{x_i}$ has exactly one non-zero base $B$ digit, at position $x_i$. For each base $B$ digit $k \geq 0$, the sum $\sum_{i=1}^n B^{x_i}$ has a base $B$ digit $c_k$ according to the number of times $k$ appears in the multiset $X$. Since $0 \leq c_k \leq n < B$ for all $k$, we know that there is no carry over in any of the base $B$ digits and therefore the sum of all base $B$ digits is exactly $n$. On the other hand, the number $B^y$ for a natural number $y$ has exactly one non-zero base $B$ digit, at position $y$, which is equal to $1$. Since $n > 1$, these numbers are not equal. $\square$

We are now ready to define our problem of interest.

**Definition 3** (CDP). *For a Countdown problem instance $\mathcal{C} = \langle I_1, O, \tau \rangle$, is there a sequence $\Theta$ that is a solution to $\mathcal{C}$?*

**Theorem 1.** *Under the standard binary encoding of integers, the CDP decision problem is NP-hard.*

*Proof.* The membership result is straightforward. We can see that there exists a polynomial witness for the CDP problem. The hardness can be shown by a polynomial reduction from the 3-PP problem.

Let $X = \{x_1, ..., x_n\}$ with $n = 3m$ and the sum equal to $mT$. Let $B = n + 1$ and $U = mT + 1$. We define the CDP instance $\mathcal{C} = \langle I_1, O, \tau \rangle$ where $I_1 = \{B^{x_1+U}, \ldots B^{x_n+U}\}$ and $\tau = mB^{T+3U}$.

Let $\Theta$ be a solution to the CDP instance above, viewed as an arithmetic expression over $I_1$. Since each $x_i$ is a natural number smaller than $T$, the solution must involve a summation over $m$ terms, each equal to $B^{T+3U}$. According to Lemma 2, each term must be a product of elements (cannot be a sum) and since $U$ is larger than any single $x_i$, it must be a product of exactly 3 elements. We therefore can collect these triplets going over the solution, giving us exactly a 3-partition. $\square$

## 5 Data Generation and Analysis

Existing literature focuses on small size instances, ranging from 4 input numbers (Gandhi et al., 2024; Yao et al., 2023) to 5 or 6 (Stojanovski et al., 2025). The generation methods start either from a given target and search for a list of numbers that can achieve that target (Gandhi et al., 2024) or start from a list of numbers and find a target (Stojanovski et al., 2025). The former approach does not scale - its computation complexity is exponential in the required input size and quickly becomes infeasible. Thus, we focus here on the latter approach, starting from a list of input numbers, we search for a target number. The method proposed in Reasoning-Gym by Stojanovski et al. (2025) simply performs a randomly chosen operation over the input numbers, in the given order. If the obtained target is not in the predefined range, the process is repeated. Our conjecture is that this results in targets that are more frequent to obtain with these numbers. In other words, the number of possible solutions to the problem is somewhat large, making it easier to find a solution. We propose a simple alternative. Given an input list of numbers (the initial state), we generate a random path from the initial state to a state with a single number $\tau_i$. We repeat it multiple times, choosing $\tau$ to be the least frequent element in $\{\tau_i\}_i$. To test our conjecture, we have generated a dataset according to Stojanovski et al. (2025), which we denote as `RG` (for Reasoning-Gym) and one according to our proposed method, denoted by `CD`, each with size ranging from 4 to 50, and 100 instances per size. Additionally, we generate a dataset according to the method of Stream-of-Search, by Gandhi et al. (2024). In this case, the instances are generated

backwards from the target by performing a breadth-first exploration, which makes the process extremely slow for larger instance sizes. We were able to generate instances of up to size 9. As before, we generated 100 instances of each size, 4 to 9. We denote the dataset by SoS (for Stream-of-Search).

Finally, we use the existing dataset of the 24 Game (Yao et al., 2023), which we denote by 24Game. All instances in the 24Game dataset are of size 4. We take the same 100 instances that are evaluated by Yao et al. (2023). All datasets and generation code are in the supplementary material. We perform a simple experiment, counting the number of solutions in these datasets using a DFS traversal. For efficiency, the algorithm is implemented in C++. Still, as the state space becomes large quite quickly (see section 4.1), we were only able to complete the traversal for instances of size up to 7 (within a reasonable time limit of 10 hours per instance). Figure 2 plots the number of solutions per instance in these three collec-

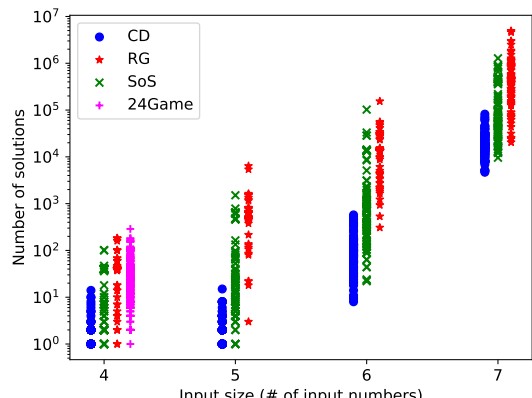

Figure 2: Solution counts, various datasets.

tions. One can clearly see that our method produces instances where the number of ways to get to the target number is significantly smaller, which arguably can indicate that these instances are harder to solve. Going back to successful benchmark desiderata mentioned in Section 2, it is clear that our proposed benchmark meets all these criteria.

## 6 EXPERIMENTAL EVALUATION

All experiments are performed on Intel(R) Xeon(R) Gold 6248 CPU @ 2.50GHz machines, with the timeout of 30 minutes and memory limit of 3.5GB per run. In all experiments, we measure accuracy in terms of the number of successfully solved instances per size. As we have 100 instances per size in each dataset, the accuracy is a number between 0 and 100. To do that, we have implemented a validator according to Definition 1. An access to a validator also allows us to measure the accuracy of a best out of k solutions produced. In our experiments that involve language models, we repeat each experiment 5 times and measure accuracy@5, choosing per task the maximal accuracy over the 5 trials. Here as well, we aggregate over the 100 instances per instance size.

### 6.1 SYMBOLIC PLANNING

We implemented a symbolic solver based on a domain-independent numeric planning. To do that, we described the Countdown problem in a planning language PDDL (Fox & Long, 2003). The PDDL domain is shown in Figure 9 in the Appendix. Each instance in our dataset is automatically translated into a PDDL problem instance. For example, an instance with input numbers $[3, 4, 5, 6]$ and a target 24 is depicted in Figure 10 in the Appendix. We use an off-the-shelf numeric planner ENHSP (Scala et al., 2020). Since the planner is deterministic, we run it only once.

### 6.2 LLM-ASSISTED PLANNING

Our evaluation focuses on the following three representative open language models: DeepSeek V3 (DeepSeek-AI et al., 2025), Llama 3.1 405B (Dubey et al., 2024), and Qwen 2.5 72B (Team, 2024). All models were accessed using API. We evaluate them in a variety of methods for planning with language models. We repeat each experiment 5 times and measure the accuracy@5, scoring 1 if at least one of the 5 attempts was successful in solving the problem.

#### AUTOTOS

We start with the most promising approach, AutoToS (Cao et al., 2024) that extends the Thought of Search framework (Katz et al., 2024). Both ToS and AutoToS achieve 100% accuracy on the related domain 24 Game. Further, these methods use the language models to produce a code that can be then used to solve all problems in the dataset with no additional calls to the language models.

This makes AutoToS a promising approach to Countdown. Our implementation of the Countdown game in AutoToS is an adaptation from the 24 Game implementation of Cao et al. (2024). We repeated the experiment 5 times, and each time, each of the tested models was able to finish the process producing the code that evaluated to 100% on the held out small set of instances. The average number of calls to the language model during AutoToS was 3.8 for DeepSeek V3, 3.4 for Llama 405B, and 4.2 for Qwen 2.5. To test the generated code, we integrated it into a standard implementation of a DFS search. As AutoToS essentially generates symbolic search-based planners, and ENHSP is a symbolic search-based planner, we can now run these planners on our dataset without using a language model.

Figure 3 depicts the accuracy of the symbolic search-based methods, ENHSP and AutoToS on our dataset. Note the interesting drop in performance between the input size 7 and 17, after which it goes back to 100%, until after size 30, when the instances become too large for the domain-independent planner ENHSP. Whenever ENHSP failed to produce a plan, it was due to a timeout - the underlying greedy best-first search (GBFS) is a heuristic search, and with increased instance size, the heuristic value computation time also increases. The simple blind DFS search, however, not needing to compute heuristic values, seems to deal rather well with large instances. Whenever it failed, it was due to exhausting the allowed memory. We note that this is due to our naive

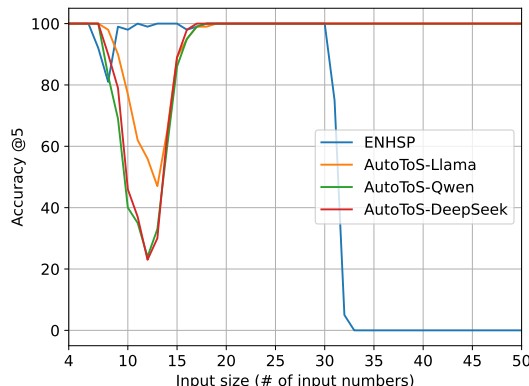

Figure 3: The accuracy of ENHSP and accuracy@5 of AutoToS with different language models for the Countdown problem.

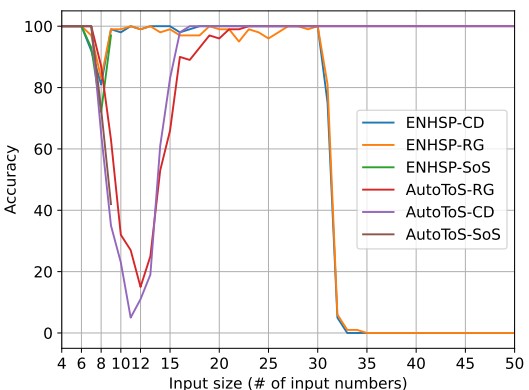

Figure 4: The accuracy of ENHSP and AutoToS for the Countdown problem, various datasets.

implementation and a different implementation of DFS might not have the memory issue. Regardless of the reasons for failure, both methods exhibit a non-monotonic performance, an unexpected phenomenon. To explore the phenomenon further, we check whether it persists on the two other mentioned datasets, RG and SoS. We choose a single AutoToS configuration, to avoid the noise from multiple trials. Figure 4 shows that the same phenomenon occurs on all tested datasets, which were created by different methods, and it happens around the same instance size values. This indicates that the Countdown game has two phase transitions, one from easy to hard around instance size 8 and one from hard to easy around instance size 20. While we cannot offer any explanation for the phenomenon, it does allow us to conclude that it is sufficient to limit our test set to sizes between 4 and 10, allowing us to capture a sufficient number of both easy and hard instances. This is not just convenient, it is necessary, as some of the LLM-based planning methods are quite computationally intensive (Katz et al., 2024).

We move now to the three popular methods of planning with language models. For simplicity, we will henceforth refer to them as LLM planning methods.

IO/CoT/ToT

The simplest and the most straightforward LLM planning method is to ask the language model to produce a solution at once, providing the problem description in the input prompt. We denote the method by (IO) for input/output. Chain of Thoughts (CoT) (Wei et al., 2022) is among the most popular methods of solving reasoning problems, eliciting the models to produce a chain of reasoning steps that lead to the final answer.

Tree of Thoughts (ToT) (Yao et al., 2023) is among the most well-cited approaches to planning with language models. The work experimented with a dataset of 24 Game instances, and therefore only a minor adaptation to their code was needed to run on our dataset.

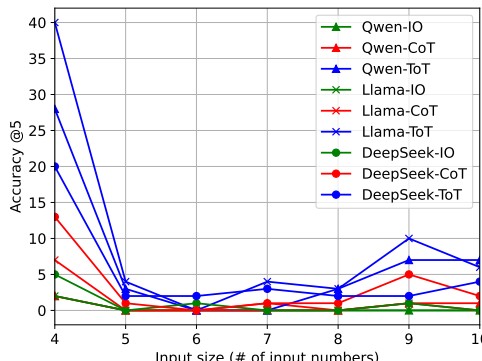

Figure 5 shows the accuracy @5 of these three LLM planning methods on our dataset. As previously mentioned, we restricted the test set to sizes between 4 and 10. Still, some methods, such as ToT, require a significant number of calls to the language model. Figure 6 presents the average number of calls to each of the language models performed while solving an instance from the CD dataset. Note that the number of calls to the language model for the IO and CoT approaches is always 1.

Figure 5: Accuracy @5 of LLM planning methods on CD.

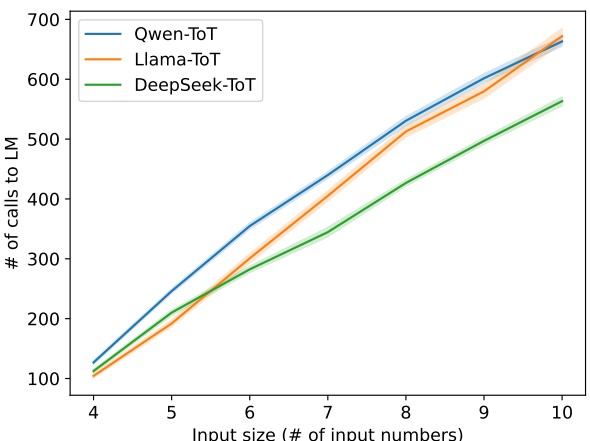

Figure 6: The average number of calls made to language models by the ToT approach with various language models.

The number of calls to the language models for the AutoToS method is below 5 for the entire dataset, regardless of the number of instances, since it is only performed once to obtain the search components code, and then no calls to a language model are made per input.

Comparing the performance result in Figure 5 to the earlier methods, depicted in Figure 3, we see a huge gap in accuracy results. The best result for LLM planning methods is 40% for input size 4, while on larger inputs all LLM planning methods score below 10%. An observant reader might notice the discrepancy from the results reported by Yao et al. (2023) on the 24 Game, 74%. While some of the difference can be attributed to the use of a different language model, GPT4, we offer an alternative explanation – some of the difference can be attributed to the way the dataset for the 24 Game was created by Yao et al. (2023). *The 24 Game instances were obtained from the internet[2], which also happens to be the source for the data used for training the language models.* In order to test this hypothesis, we ran the three LLM planning approaches on the instances from Yao et al. (2023), depicted by `24Game`.

Figure 7 and Table 1 show the comparison between accuracy obtained on `24Game` and instances of size 4 in our dataset `CD[4]`. The figure visualizes the accuracy @5 results while the table presents the raw numbers for both the accuracy @5 and the mean accuracy. For each of the models and each of the methods, we can clearly observe the significant drop in accuracy when moving away from the instances the models might have seen in their training data.

|  | Model | IO 24Game | IO CD[4] | CoT 24Game | CoT CD[4] | ToT 24Game | ToT CD[4] |
|---|---|---|---|---|---|---|---|
| acc@5 | Qwen | 6 | 2 | 8 | 2 | 83 | 28 |
| | Llama | 7 | 2 | 32 | 7 | 90 | 40 |
| | DeepSeek | 38 | 5 | 48 | 13 | 77 | 20 |
| mean | Qwen | 2 | 1 | 2 | 0 | 47 | 9 |
| | Llama | 1 | 0 | 9 | 1 | 48 | 12 |
| | DeepSeek | 10 | 1 | 18 | 4 | 28 | 4 |

Table 1: Accuracy of the LLM planning methods.

---

[2]https://www.4nums.com/game/difficulties/

This gives a strong indication for the utility of the proposed data generation method and the CD dataset and its superiority over the existing datasets. It is worth mentioning that the time to generate a single game instance is typically short. With 100,000 iterations (the setting we used), it takes from 3 seconds for size 4 to 50 seconds for size 50, and it depends linearly on the number of iterations. Since we propose a generation method that can easily and quickly produce previously unseen data, we do not have the disadvantage of static datasets that gradually find their way into the training sets of language models.

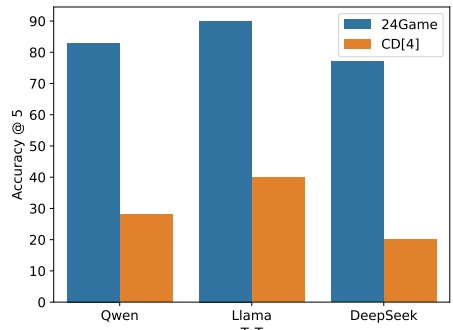

Figure 7: Accuracy @5 of various LMs with Tree of Thought (ToT), 24Game dataset vs. instances of the same size (4) from our dataset.

IO/CoT WITH REASONING MODELS

Reasoning models seek to mitigate the limitations of standard language models by making their reasoning process explicit. Their compute heavy nature make them less suitable to approaches that require many calls to the model per question. While being more compute-intensive, they often can compensate in quality in other cases. We therefore evaluate the performance of reasoning models for IO and CoT querying methods. We test three open reasoning models: Qwen3-30B-A3B-Thinking-2507 , DeepSeek R1 , and GPT-OSS-120B . It is worth mentioning that we had to significantly increase the maximal allowed tokens bound (from 1k to 6k) to achieve non-zero performance for these models. Table 2 shows mean accuracy across 5 runs, comparing reasoning and pure language models. Reasoning models indeed take significantly more effort than regular language models, but do exhibit somewhat better performance as can be seen by comparing Qwen2.5 to Qwen3. We note that Deepseek R1 frequently times out even for IO on tasks of size 4 and that GPT-OSS-120B frequently exceeds the tokens bound.

|  | Model | 4 | 5 | 6 | 7 | 8 | 9 | 10 |
|---|---|---|---|---|---|---|---|---|
| IO | Qwen2.5 | 0.6 | 0 | 0 | 0 | 0 | 0 | 0 |
| | Llama | 0.4 | 0 | 0 | 0 | 0 | 0 | 0 |
| | DeepSeek V3 | 1.4 | 0 | 0.2 | 0 | 0 | 0.2 | 0 |
| | Qwen3 | 32.0 | 3.4 | 1.8 | 1.6 | 4 | 1.2 | 0.8 |
| | GPT-OSS | 25.6 | 1.0 | 0.8 | 1.2 | 3.4 | 2.0 | 3.2 |
| | DeepSeek R1 | 23.0 | 1.0 | 0 | 0 | 0 | 0 | 0 |
| CoT | Qwen2.5 | 0.4 | 0 | 0 | 0.2 | 0 | 0.2 | 0.2 |
| | Llama | 1.4 | 0 | 0 | 0 | 0 | 0.2 | 0 |
| | DeepSeek V3 | 4.0 | 0.2 | 0 | 0.2 | 0.2 | 1.0 | 0.4 |
| | Qwen3 | 42.6 | 4.6 | 2.6 | 3.2 | 6.8 | 5.4 | 4.6 |
| | GPT-OSS | 22.4 | 1.0 | 0 | 0 | 1.0 | 0.8 | 0 |
| | DeepSeek R1 | 25.0 | 1.0 | 0 | 0 | 0 | 0 | 0 |
| ToT | Qwen2.5 | 9.2 | 0.6 | 0 | 0 | 2.0 | 2.8 | 5.0 |
| | Llama | 12.2 | 0.8 | 0 | 2.4 | 2.2 | 5.0 | 4.0 |
| | DeepSeek V3 | 4.5 | 0.5 | 0.4 | 0.8 | 0.4 | 0.6 | 1.6 |

Table 2: Mean accuracy of the LLM/LRM planning methods.

## 7 ERROR CLASSIFICATION AND ANALYSIS

To better understand the errors made by the language models, we partition them into categories:

- *Incorrect Format*, where the output generated didn't align with the format that was specified in our prompt.

- *Less Number of Steps Used*, where the number of steps used in the solution identified by the planner was smaller than the required number of steps, which should always be equal to the size of the input numbers.

- *More Number of Steps Used*, where the number of steps is longer than what is required. Note that all valid solutions for a given countdown problem have exactly the size of the input numbers minus one operations.

- *Not All Input Numbers Used*, where one or more of the input numbers were not used along the provided solution.

- *Not Target Number*, where the sequence of operations listed in the solution results in a number different from the target number.

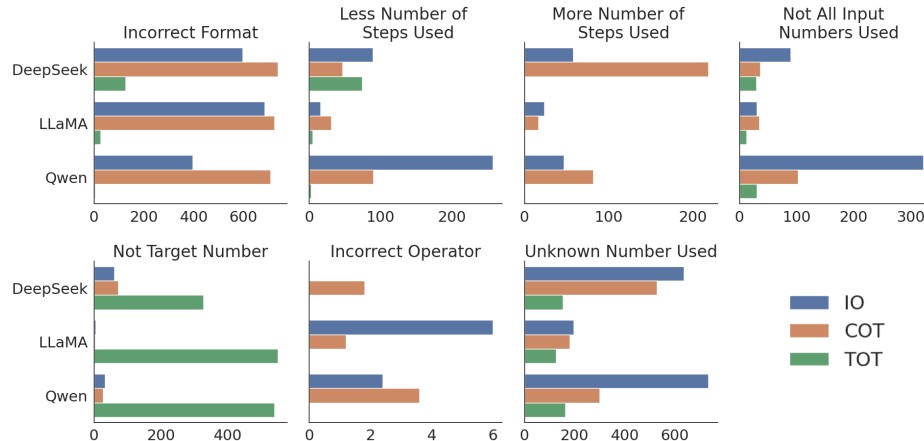

Figure 8: Mean number of error observed per language model and planning method across each each error category.

- *Incorrect Operator*, where the operator sequence uses an operator outside the set of operators $O$ considered in this version of the countdown problem.

- *Unknown Number Used*, where a solution step mentions a number that should not be available at that step.

Note that these errors are not disjoint, sometimes multiple errors appear at the same solution step. Figure 8 shows the mean of the frequency of error observed by various methods with different models, across 5 runs. Note that the figure includes only IO, CoT, and ToT methods, since all solutions produced by AutoToS were validated to be correct. The baseline, ENHSP is guaranteed to only generate correct solutions, as the planning model is correct (human validated) and the planner is both sound and complete. Observe that per method (IO/CoT/ToT), with just a few exceptions, the models are not too different in the errors they make.

The three most common categories, responsible for the lion share of all errors are formatting errors, use of unknown number, and reaching a number different from the target one. ToT seems to exacerbate the issue with the latter two categories, which together are responsible for 67.7%, 94.1%, and 95% of all errors of DeepSeek, Llama, and Qwen, respectively. Incorrect operators are by far the rarest category, with no such errors in ToT. Next two are the more/less than needed number of steps, with similar share of errors falling into these two categories. Finally, not all input numbers being used appears mostly in IO, sometimes in CoT, rarely in ToT.

## 8 CONCLUSIONS AND FUTURE WORK

We make a case for the Countdown game as a benchmark of models and agents' planning abilities. This easily describable in natural language yet precise and computationally challenging domain meets many desiderata of an ideal planning domain. We compare the performance of various LLM-assisted planning methods as well a symbolic baseline based on a domain-independent numeric planner and find AutoToS to perform best overall, while the famous LLM-based planning methods IO, CoT, and ToT exhibit inadequate performance (below 10%) for instance sizes larger than $4$. Further, even for instances of size $4$, the performance of these methods drops dramatically compared to the performance on the static dataset from their original experimental evaluation. This raises serious concerns about the suitability of these methods for solving previously unseen planning problems.

In future, we would like to explore various extensions of Countdown. Allowing additional operations or using only a subset of input numbers might have a positive effect on language models' performance. On the other hand, introducing different costs of operations and optimizing the summed cost of a sequence makes the problem harder, and will challenge the currently well performing methods.

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

APPENDIX

## A  COUNTDOWN PDDLs

The PDDL domain file for the proposed countdown domain, used in our experiments with ENHSP symbolic planner is provided in Figure 9. Figure 10 provides an example instance with input numbers [3,4,5,6] and a target 24.

```
(define (domain countdown)
  (:types num - object)
  (:predicates (active ?o-num) (goalreached))

  (:functions (value ?o - num) (targetvalue) (numactive))

  (:action add
   :parameters (?a ?b - num)
   :precondition (and (not (= ?a ?b)) (active ?a) (active ?b))
   :effect        (and (decrease (numactive) 1)
                      (increase (value ?a) (value ?b))
                      (not (active ?b))))

  (:action subtract
   :parameters (?a ?b - num)
   :precondition (and (not (= ?a ?b)) (active ?a) (active ?b)
                      (>= (value ?a) (value ?b)))
   :effect        (and (not (active ?b)) (decrease (numactive) 1)
                      (decrease (value ?a) (value ?b))))

  (:action multiply
   :parameters (?a ?b - num)
   :precondition (and (not (= ?a ?b)) (active ?a) (active ?b))
   :effect        (and (not (active ?b)) (decrease (numactive) 1)
                      (assign (value ?a) (* (value ?a) (value ?b)))))

  (:action divide
   :parameters (?a ?b - num)
   :precondition (and (> (value ?b) 0) (not (= ?a ?b))
                      (active ?a) (active ?b))
   :effect        (and (not (active ?b)) (decrease (numactive) 1)
                      (assign (value ?a) (/ (value ?a)(value ?b)))))

  (:action checkgoal
   :parameters (?a - num)
   :precondition (and (active ?a) (= (numactive) 1)
                      (= (value ?a) (targetvalue)))
   :effect        (and (goalreached)))
)
```

Figure 9:  The PDDL domain for the Countdown problem.

```
(define (problem c01)
  (:domain countdown)
    (:objects n1 n2 n3 n4 - num)
  (:init
    (= (value n1) 3) (= (value n2) 4) (= (value n3) 5) (= (value n4) 6)
    (= (targetvalue) 24)
    (= (numactive) 4)
    (active n1) (active n2) (active n3) (active n4)
  )
  (:goal (and (goalreached)))
)
```

Figure 10:  The PDDL problem example for input [3,4,5,6] and target 24.

## B   COMPLETE PROOF OF LEMMA 3

**Lemma 2.** *There exist no two sets of integers $\{x, y\}$ and $\{a, b\}$, such that*

$$10^{a \pm b} = 10^x \pm 10^y$$

*Proof.* **Case 1:** $10^{a+b} = 10^x + 10^y$  proof is in the main part of the paper.

**Case 2:** $10^{a-b} = 10^x - 10^y$

Assume to the contrary that $a, b, x, y \in \mathbb{N}$ such that $10^{a-b} = 10^x - 10^y$. Then $a - b = \log(10^x - 10^y)$ and therefore $\log(10^x - 10^y) \in \mathbb{N}$.

Assume w.l.o.g that $x > y$. Observe that $\log(10^x - 10^y) = \log(10^y 10^{x-y} - 10^y) = y + \log(10^{x-y} - 1)$. Therefore, $\log(10^n - 1) = m \in \mathbb{N}$ for some $n \in \mathbb{N}$. Thus, $10^m = 10^n - 1$ or $1 = 10^n - 10^m$. Since $f(x) = 10^x$ is monotonically increasing, this can happen only when $n > m$. Since $m, n \in \mathbb{N}$, this means that $n \geq m + 1$. Therefore we have

$$1 = 10^n - 10^m \geq 10^{m+1} - 10^m = (10 - 1)10^m > 1 \cdot 1,$$

contradicting the assumption.

$\square$

