# OpenReview forum: "Seemingly Simple Planning Problems are Computationally Challenging: The Countdown Game"
_ICLR.cc/2026/Conference — Submitted to ICLR 2026_

### Official Review · Reviewer_mjhu · 2025-10-23

**Soundness:** 3
**Presentation:** 4
**Contribution:** 3
**Rating:** 10
**Confidence:** 3

**Summary:**

This is a strong, interesting paper that investigates the planning capabilities of today's LLM. It proposes a simple, easy-to-use benchmark for assessing LLM planning, and it discusses the empirical results.

**Strengths:**

The work appears to be original. The paper is of high quality and clarity, and the empirical results show that the proposed benchmark allows us to better understand the limitations of today's LLM-based planning.

**Weaknesses:**

Even though the proposed benchmark is clearly valuable, the paper would greatly benefit for an extensive discussion of its current limitations and possible improvements. Such a discussion would be an excellent roadmap for future work on this area.

**Questions:**

N/A

---

> ### Author Response · Authors · 2025-11-19
>
> We thank you for your support!

---

### Official Review · Reviewer_44jd · 2025-10-30

**Soundness:** 3
**Presentation:** 4
**Contribution:** 3
**Rating:** 6
**Confidence:** 2

**Summary:**

This paper proposes Countdown as a planning benchmark with an NP-completeness proof, an instance generator, and a Planning Domain Definition Language (PDDL) model for planner baselines. The authors evaluate multiple LLM methods including Input-Output,  Chain-of-Thought, Tree-of-Thoughts,, and AutoToS. They find an easy-hard-easy difficulty pattern across instance sizes. Dynamic instances reduce ToT performance relative to the static 24 Game dataset, suggesting memorization rather than generalization.

**Strengths:**

Shows that prompting methods may lack generalization on unseen instances. Domain formulation enables reproducible evaluation. Strong baselines including symbolic planners.

**Weaknesses:**

Unlcear how presentative the Countdown problem actually is.

**Questions:**

Could you provide diagnostic data like branching factors, node expansions, or solution multiplicity distributions across sizes?
The NP-completeness proof needs clarification on numeric encoding.

---

> ### Author Response · Authors · 2025-11-19
>
> Thank you for your constructive feedback and for raising these questions.
>
> The mathematical computation of the branching factor bound, as well as number of states per layer and the total number of states are presented in Section 4.1. The branching factor in a state of size k is upper-bounded by $b_k = k * (k − 1) * 3$.  The computed numeric bound on the state space size per input size is depicted in Figure 1.
>
> The statistical data on the number of solutions in the generated instances is presented in Figure 2.
>
> We will update the paper to clarify on numeric encoding.

---

> ### Author Response · Authors · 2025-11-26
>
> We have updated the theorem and the proof, including clarification on the encoding.

---

### Official Review · Reviewer_dWGU · 2025-10-31

**Soundness:** 2
**Presentation:** 3
**Contribution:** 2
**Rating:** 4
**Confidence:** 3

**Summary:**

The paper proposes a procedure for creating a planning benchmark based on the Countdown game. In this game, a player is expected to form a target number from a list of input numbers through arithmetic operations. It is shown that the problem is NP-complete, thus it is computationally challenging. The paper argues that the domain meets many desiderata associated with an ideal benchmark for planning capabilities evaluation. It includes an extensive evaluation of its instance generation procedure over public benchmarks. The conclusions drawn from the experiments are interesting.

The paper lists several desired properties for a benchmark of planning abilities. It would have been helpful if the paper elaborates on these criteria (e.g., why is a criterium desirable?). Furthermore, there is no discussion that shows that the generated benchmark satisfies these desired properties. In my opinion, the fact that the instances are planning problem instances does not provide an obvious answer, for otherwise, any procedure that generates planning problem instances could be used.

The presentation is easy to follow. However, some definitions and proofs need to be clarified:

- Definition 3: do all numbers in X need to be used to construct $\omega$? For example, does the SAP problem with X = {4, 5} and $\omega=4$ have a solution?
-  I am not sure if Lemma 3 is sufficient to conclude that the CDP constructed in Theorem 1 cannot contain + and -. What if there are a, b, c and x, y, z so that $10^{a \pm b \pm c} = 10^x \pm  10^y \pm  10^z$? (e.g., if a = x, b = c, y=z  then $10^{a +b - c} = 10^x +  10^y - 10^z$). So, it appears that Lemma needs to be proved for two arbitrary sets $\{x_1, \ldots, x_k\}$ and $\{y_1, \ldots, y_k\}$. This is, however, not correct for k = 3.

The description of the generation of the dataset is really good.

The experiment section is extensive. It demonstrates that the generated instances are difficult for LLM-based planning.

Overall, the paper proposes a new benchmark for planning. There is some disconnection between parts of the papers and a proof might need to be revised.

**Strengths:**

+ The paper is easy to follow.

+ When comparing the performance of LLM planning methods on the 24Game instances and the CD dataset generated by the proposed method, all LLM planning methods show significantly worse performance on the CD dataset. The finding indicates the usefulness of the proposed data generator.

+ The manuscript clearly explains the setting and the metric. Aggregating 100 instances per instance size sees a solid design.

+ Three representative open language models are used. They are a good representation of LLMs.

**Weaknesses:**

- Please check my comments on the proof of Theorem 1.

- The discussion on desired properties of a planning benchmark should be discussed and connected with other parts of the paper.

- The data generation does not report the time used to generate CD, RG, and SoS. It will be good to include the running time.

- To show the superiority of the CD dataset, generated by the proposed method, it will be better to run the LLM planning methods on the CD datasets and other generated datasets, RG, and SoS. However, the paper only compares the performance of the LLM planning methods on the CD and 24Game datasets.

**Questions:**

- Please check my comments on the desired properties and the proof of Theorem 1. They need to be addressed.

Other questions:
- The measure accuracy @5 chooses per task the maximal accuracy over the 5 trials. Why do not report the average accuracy over the 5 trials?

- It will be useful to investigate further the fundamental reason in the dip of Figures 2 &3.

---

> ### Author Response · Authors · 2025-11-19
>
> Thank you for your constructive feedback and for raising these questions. We try to clarify below in the hopes of alleviating your concerns. We are very open to discussion!
>
> **On Desiderada**: We felt that the reasons for these desired properties are self evident. The only exception where it might have been non-self evident is the last property, where we add a reason. We briefly specify in the conclusions that Countdown meets the desiderada. It is quite obvious and we felt that the allowed space can have a better use. We can add specific references to the desiderada items throughout the paper.
>
> **On Definition 3**
> You are correct, in Definition 3 all elements of X must be used.  We will adapt the text to make it absolutely clear.
>
> **Re Lemma 3 and triplets**:
> According to Def. 1, a solution is a sequence of triplets, where each triplet <$x_i, o_i, y_i$> involves two numbers and an operation. Each step must be valid for the solution to be valid. Hence, Lemma 3 is sufficient, as there are no steps that involve more than two numbers.
>
> **On experiments with other datasets**: Figure 2 shows the comparison between the known methods of generating instances in terms of the number of solutions.  Figure 4 shows the performance of ENHSP and AutoToS (LLM-assisted method) on these datasets. The existing literature have already shown the performance of pure LLM-based methods on their datasets. Our aim was not to present a superior dataset, but to better analyze the Countdown domain. The improvement of the existing generation methods was not the main purpose of this work.
>
> **The instance generation time**: For SoS, as we mention in the paper, the process is extremely slow. We were not able to generate instances of size 10 within our time and memory limits (30 mins, 3.5GB). For ReasoningGym (RG), the generator is fast for smaller instances (less than a second for sizes up to 12), but becomes slow rather quickly. Generating an instance of size 30 takes a minute and of size 50 takes 10 minutes. For our dataset (CD), with 100,000 iterations (the setting we used), it takes from 3 seconds for size 4 to 50 seconds for size 50.  We will add these to the paper.
>
> **On accuracy@5**: For deterministic methods (ENHSP, AutoToS with learned successor function and goal test), Figure 4 shows the accuracy of a single run.
> For the LLM-based methods, Table 1 shows both accuracy@5 and mean results, for instances of size 4 (columns CD[4]). For example, ToT with Llama achieves accuracy@5 of 40, and mean value of 12.

---

> ### Comment · Reviewer_dWGU · 2025-11-26
> **Lemma 3**
>
> Thanks for answering my questions. I still have some difficulty with your explanation related to Lemma 3. Please see the following.
>
> Suppose that you have the SAP instance X = {2, 3, 3} and $\omega =2$.
>
> The corresponding Countdown problem is $I_1$ = {$10^2, 10^3, 10^3$} with $\tau = 10^2$.
>
> You wrote that "According to Lemma 3, a solution to this Countdown problem cannot contain + or − operations." but there is a solution containing + and -, which is $10^2 +  10^3 - 10^3$.
>
> Am I missing something?

---

> > ### Author Response · Authors · 2025-11-26
> >
> > You are correct, Lemma 3 makes an assumption that the elements differ and therefore is not a good fit for the complexity proof in our case, when dealing with multisets. The proof had another issue of representation size, so we have been working on a different proof that overcomes the issue. We have now updated the paper with the new proof, reducing from the strongly NP-complete 3-partitioning. The 3-partitioning problem deals with summation of natural numbers and does not have the issue you mention. We apologize for the oversight and hope the reviewer could review the new version of the proof.

---

### Official Review · Reviewer_wHms · 2025-10-31

**Soundness:** 3
**Presentation:** 2
**Contribution:** 2
**Rating:** 4
**Confidence:** 3

**Summary:**

The paper focus on the drawback of existing planning benchmarks, such as poorly defined tasks or benchmark memorization.
 To address this, the authors propose using the countdown game as a superior benchmark.
 An experimental evaluation showing that while traditional symbolic planners and an LLM-based solver-generator can succeed, popular LLM planning methods fail dramatically on these new instances, even those of small size. This suggests their high performance on existing benchmarks may be due to data contamination rather than genuine planning ability.

**Strengths:**

1. The paper provides a formal proof that the countdown problem is NP-complete, which gives a strong theoretical foundation for its use as a hard benchmark, moving beyond qualitative assessments.
2. The authors' proposed a automatic data generation method, which favors targets with fewer solution paths, directly tackles the problem of benchmark memorization.

**Weaknesses:**

1.  The paper fails to explain why there is an easy to hard to easy situation in Figure 3&4, undermining the complexity of such tasks.
2. The error analysis categorizes failure modes but doesn't provide a deep qualitative analysis of why these errors occur. It's unclear if LLMs fail at basic arithmetic, logical step-tracking, or strategic search.
3. The dataset is constructed in an over-simplified manner. Although it does evaluate the core planning ability, its over-simplification undermines its application range.

**Questions:**

1. Given that direct planning methods fail, while the successful AutoToS method that use the LLM to generate symbolic code for a traditional search algorithm, does this suggest that the most viable role for LLMs in complex, verifiable planning is not as direct planners but rather as problem-compilers that translate natural language descriptions into a formal, symbolic representation that a dedicated solver can then execute?
2. How can the benchmark be used to improve the performance of the base LLM?

---

> ### Author Response · Authors · 2025-11-19
>
> Thank you for your constructive feedback and for raising these questions. We try to clarify below in the hopes of alleviating your concerns. We are very open to discussion!
>
> **Q1**. Planning models generation, either in code, as is done by AutoToS, or directly in high-level planning languages such as PDDL is indeed a promising direction for language models. In mission-critical tasks, where there is a need for guarantees of soundness, such models present the opportunity to validate the solver, avoiding the need to validate each generated solution.
> However, this is merely one possible direction to follow. We envision many possible hybrid approaches that use LLMs to decompose the planning task, to generate complex, reusable multi-step macros, to generate search guidance heuristics, etc. We are happy to see that some of these ideas are already tackled in recent work. We hope that our benchmark will allow to correctly evaluate the planning abilities of these future approaches.
>
> **Q2**. As to how the benchmark can be used to improve the performance of base LLMs, arguably, it shouldn't.
> Training LLMs to perform NP-hard tasks in a particular domain is only sensible in practically useful domains, not in benchmarks that intend to measure progress in certain capabilities. Solving a benchmark should not be a goal by itself, but rather a measure of progress towards the goal.

---

### Official Review · Reviewer_uTgf · 2025-11-01

**Soundness:** 3
**Presentation:** 4
**Contribution:** 4
**Rating:** 8
**Confidence:** 4

**Summary:**

This paper introduces Countdown Game as a new benchmark for evaluating long-horizon planning in LLMs. Unlike prior loosely defined or classical planning tasks, it formalizes the arithmetic-based game as a planning problem and proves it NP-complete via reduction from Partition. The authors develop a diverse instance generator, a PDDL formulation, and evaluate both classical planners and LLM-based methods. Results show Countdown is significantly harder than simpler tasks like the 24-Game, exposing key failure modes in current LLM planners. The work provides a rigorous, reproducible benchmark and insights into the limitations of LLM reasoning.

**Strengths:**

- The paper provides a solid theoretical foundation by formally analyzing the problem’s complexity.
- The authors design and validate a method to generate challenging and diverse puzzle instances, addressing potential issues of triviality or memorization. Instead of relying on fixed or scraped puzzles which could appear in training data, they propose a dynamic instance generator.
- The paper evaluates a broad range of planning approaches under consistent conditions, yielding credible and insightful results. The authors test multiple paradigms: a classical planner, a cutting-edge neuro-symbolic method, and popular prompt-based LLM strategies, making the evaluation well-rounded.
- The paper surfaces important observations that advance understanding of LLM planning. Two discoveries stand out: (a) a phase transition phenomenon in puzzle difficulty, and (b) evidence of overfitting on static benchmarks. The paper also delves into why the LLM planners fail, which is very useful for future improvements.

**Weaknesses:**

- The paper reports a surprising “hard-to-easy” phase transition in problem difficulty without providing an explanation, leaving a gap in understanding. While the discovery of two phase transitions is intriguing, the authors explicitly note that they cannot offer an explanation for why larger Countdown instances become easier again.
- The paper primarily compares LLM-based planners against one classical planner and each other, but other potential baseline methods are not explored. For instance, Countdown puzzles could be approached with a brute-force or heuristic search specialized for arithmetic outside of PDDL, or even formulated as a Constraint Satisfaction/Optimization problem (e.g., an ILP or backtracking solver). Including a straightforward depth-first search or an optimized backtracking solver for small instances would show how far pure symbolic search can go on these problems.

**Questions:**

- What might be the underlying cause of the second “hard-to-easy” phase transition around 20 input numbers, and can the authors provide any analysis or hypotheses here?
- Have the authors considered evaluating alternative solving approaches, such as a bespoke backtracking or integer programming solver for the Countdown puzzle, or integrating a planning heuristic beyond ENHSP’s capabilities?

---

> ### Author Response · Authors · 2025-11-19
>
> Thank you for your constructive feedback and for raising these questions.
>
> **Q1**. The “hard-to-easy” phase transition still puzzles us. Some hypotheses we had are that at some point a zero can be derived and that allows not to use all numbers. We welcome a discourse on this topic. In fact, our hope is that our work will give rise to such a discourse.
>
> **Q2**
> 1. The existing baseline, ENHSP, performs rather well. We did not optimize its hyper-parameters to Countdown, which could result in even better performance. ENHSP in the default configuration performs greedy best-first search with a heuristic function based on delete relaxation.
> The AutoToS-based methods use a simple straightforward implemenation of depth-first search. Nevertheless, these approaches are already showing quite strong performance.
> 2. It would be an interesting and very challenging task to create a mixed integer linear program for Countdown. Multiplication and division are non-linear operations, making the task non-straightforward. We would encourage the operations research community to tackle this problem!

---

### Official Review · Reviewer_y4Za · 2025-11-02

**Soundness:** 1
**Presentation:** 2
**Contribution:** 1
**Rating:** 0
**Confidence:** 4

**Summary:**

This paper focus on the problem of countdown game, provided a theoretical analysis of countdown game to show that it is NP-hard. The problem is evaluated with some LLMs with existing LLM-based planning approach like CoT, ToT, and AutoTos to show that the proposed problem is still a challenging problem.

**Strengths:**

- The paper provides a theoretical guarantee of the proposed problem.
- The paper is overall well-written and easy to follow.

**Weaknesses:**

- The paper does not follow the paper format.
- As the author acknowledged, the same problem has been previously proposed in Reasoning Gym [1].
- The evaluation is based on old models that are even reasoning models. As a dataset paper, this is highly insufficient. To the current stage of LLM research, it is no surprise that LLM will fail in some NP-hard problem, but it will be much more interesting if one can provide more insights into what scale wat ill the problem becomes unsolvable to the latest models, and how potentially it can be improved. The current contribution is far below what is expected from an ICLR paper. The current paper is overall much less informative than its prior work [1].

[1]. Stojanovski Z, Stanley O, Sharratt J, et al. REASONING GYM: Reasoning Environments for Reinforcement Learning with Verifiable Rewards[J]. arXiv preprint arXiv:2505.24760, 2025.

**Questions:**

No specific questions. While this paper is probably a technically correct paper, I believe the paper still needs substantial improvement to be accepted to a top-tier ML conference.

---

> ### Author Response · Authors · 2025-11-24
>
> **On the connection to Reasoning Gym**:
> We have tried to the best of our abilities to trace the origins of the Countdown game, which we cite in the paper. Reasoning Gym indeed also proposes a dataset for the Countdown game, among many others. Their paper however does not suggest any insights about Countdown. In fact, Countdown is not even mentioned in the main paper, only in the appendix. According to the appendix, the instances they generate are of sizes between 4 and 6, and, as we argue in the paper, the method used results in very easy to solve instances, which is supported by their experiments. This is also supported by Figure 2 in our paper, showing that their instances have several orders of magnitude more solutions.
> To sum up, we believe that this work presents significantly more insights into the Countdown benchmark than the existing literature.
>
>
> **On reasoning models**:
> Reasoning models are extremely computationally expensive and therefore are not a good fit for methods such as ToT, where hundreds of calls are performed per question.
> For IO/CoT, we tested open reasoning models Qwen3-30B-A3B-Thinking-2507, DeepSeek R1, and GPT-OSS-120B. We had to significantly increase the maximal allowed tokens bound (from 1k to 6k) to achieve non-zero performance.
>
> While taking much more time than Qwen2.5, Qwen3 achieves somewhat better performance. Deepseek R1 frequently times out even for IO on tasks of size 4. GPT-OSS-120B frequently exceeds the tokens bound.
>
> Specifically, the accuracy@5 values are:
> | Model | Method   | 4 | 5 | 6 | 7 | 8 | 9 | 10 |
> | :------- | :------- | :------: | -------: | -------: | -------: | -------: | -------: | -------: |
> |Qwen3| IO       | 55| 5 | 7 | 7 | 14| 4 | 3 |
> |Qwen3| CoT      | 61| 10| 9 | 13| 21|22 | 18|
> |DeepSeek R1| IO       | 23 | 1 | 0 | 0 | 0 | 0 | 0 |
> |DeepSeek R1| CoT      | 25 | 1 | 0 | 0 | 0 | 0 | 0 |
> |GPT-OSS-120B| IO       | 28 | 1 | 1 | 2 | 7 | 4 | 8 |
> |GPT-OSS-120B| CoT      | 23 | 1 | 0 | 0 | 1 | 1 | 0 |
>
> For completeness, we have performed a very limited evaluation of these 3 models on ToT and did not observe any question solved correctly.
>
>
> **On latest models**:
> If by latest models the reviewer means proprietary models (such as openai GPT5, o4, etc), we respectfully disagree.
> We deliberately chose to use only open language models in our experiments to ensure reproducibility, transparency, and scientific rigor. Open models allow full access to architecture, weights, and training methodology, which is essential for peer verification and long-term reproducibility. In contrast, closed models often change without notice and provide limited visibility into training data or fine-tuning processes, making it difficult to replicate results or audit biases.
> Moreover, our approach aligns with open science principles and FAIR guidelines (Findable, Accessible, Interoperable, Reusable). By using open LLMs, we enable other researchers to reproduce our experiments without licensing or API restrictions, fostering broader participation and reducing dependency on proprietary systems. This choice ensures that our findings remain verifiable and sustainable over time.
>
> Given the above, we respectfully ask you to reconsider the score. Penalizing for prioritizing open science risks discouraging verifiable research.

---

### Meta-Review · Area_Chair_J9yM · 2026-01-08

**Summary:**

1. The game of CountDown (CD) has been introduced in previous works (e.g., Reasoning Gym) and the work is not novel
2. No evaluation on reasoning models or SoTA close models.
3. Comparison with other planning techniques (e.g., brute-force, heuristic search, mixed integer linear programming, etc)
4. The hard-easy phase transition is counter intuitive and not analyzed clearly.

**Reviewer Concerns:**

Overall the authors have addressed some issues but not all of them. The rebuttal is quite brief. Lack of evaluation on reasoning models and SoTA models is a key issue and the authors only provide initial experiments with ToT. AC advises the author to perform more extensive evaluation in the next revision.

**Reviewer Scores:**

y4Za: 0->0 (likely won't change the opinion, finding key issues)
uTgf: 8->6 (may remain positive, most of the concerns are not addressed)
wHms: 4->4 (issues not addressed well, rebuttal is brief)
dWGU: 4->6 (the proof's issue is partially addressed)
44jd: 6->6 (review is very brief, no response after the former AC asked clarification questions.)
mjhu: 10->10 (super positive but not fully justified)

---

### Decision · Program_Chairs · 2026-01-26

Reject